# Embedding Symbolic Knowledge into Deep Networks

**Yaqi Xie,** * **Ziwei Xu,** * **Mohan S Kankanhalli, Kuldeep S. Meel, Harold Soh**
School of Computing
National University of Singapore
`{yaqixie, ziwei-xu, mohan, meel, harold}@comp.nus.edu.sg`

## Abstract

In this work, we aim to leverage prior symbolic knowledge to improve the performance of deep models. We propose a graph embedding network that projects propositional formulae (and assignments) onto a manifold via an augmented Graph Convolutional Network (GCN). To generate semantically-faithful embeddings, we develop techniques to recognize node heterogeneity, and semantic regularization that incorporate structural constraints into the embedding. Experiments show that our approach improves the performance of models trained to perform entailment checking and visual relation prediction. Interestingly, we observe a connection between the tractability of the propositional theory representation and the ease of embedding. Future exploration of this connection may elucidate the relationship between knowledge compilation and vector representation learning.

## 1 Introduction

The recent advances in design and training methodology of deep neural networks [1] have led to widespread application of machine learning in diverse domains such as medical image classification [2] and game-playing [3]. Although demonstrably effective on a variety of tasks, deep NNs have voracious appetites; obtaining a good model typically requires large amounts of labelled data, even when the learnt concepts could be described succinctly in symbolic representation. As a result, there has been a surge of interest in techniques that combine symbolic and neural reasoning [4] including a diverse set of approaches to inject existing prior domain knowledge into NNs, e.g., via knowledge distillation [5], probabilistic priors [6], or auxiliary losses [7]. However, doing so in a scalable and effective manner remains a challenging open problem. One particularly promising approach is through learned embeddings, i.e., real-vector representations of prior knowledge, that can be easily processed by NNs [8, 9, 10, 11, 12].

In this work, we focus on embedding symbolic knowledge expressed as *logical rules*. In sharp contrast to connectionist NN structures, logical formulae are explainable, compositional, and can be explicitly derived from human knowledge. Inspired by insights from the knowledge representation community, this paper investigates embedding alternative representation languages to improve the performance of deep networks. To this end, we focus on two languages: Conjunctive Normal Form (CNF) and decision-Deterministic Decomposable Negation Normal Form (d-DNNF) [13, 14]. Every Boolean formula can be succinctly represented in CNF, but CNF is intractable for most queries of interest such as satisfiability. On the other hand, representation of Boolean formula in d-DNNF may lead to exponential size blowup, but d-DNNF is tractable (polytime) for most queries such as satisfiability, counting, enumeration and the like [14].

In comparison to prior work that treat logical formulae as symbol sequences, CNF and d-DNNF formulae are naturally viewed as graphs structures. Thus, we utilize recent Graph Convolutional Networks (GCNs) [15] (that are robust to relabelling of nodes) to embed logic graphs. We further employ

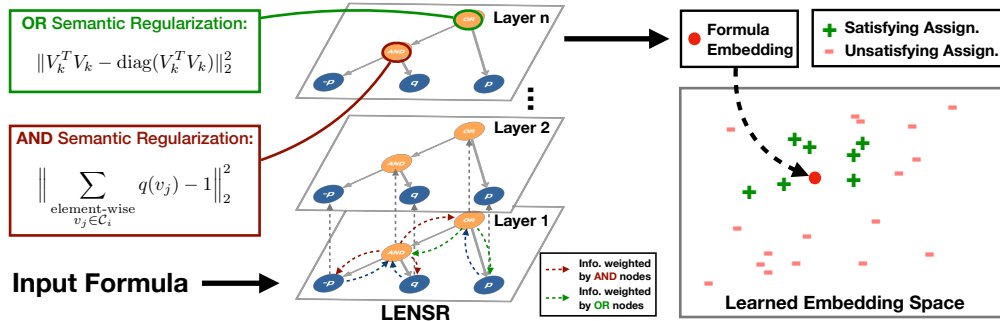

Figure 1: LENSR overview. Our GCN-based embedder projects logic graphs representing formulae or assignments onto a manifold where entailment is related to distance; satisfying assignments are closer to the associated formula. Such a space enables fast approximate entailment checks — we use this embedding space to form logic losses that regularize deep neural networks for a target task.

a novel method of *semantic regularization* to learn embeddings that are semantically consistent with d-DNNF formulae. In particular, we augment the standard GCN to recognize node heterogeneity and introduce soft constraints on the embedding structure of the children of AND and OR nodes within the logic graph. An overview of our **L**ogic **E**mbedding **N**etwork with **S**emantic **R**egularization (LENSR) is shown in Fig 1.

Once learnt, these logic embeddings can then be used to form a *logic loss* that guides NN training; the loss encourages the NN to be consistent with prior knowledge. Experiments on a synthetic model-checking dataset show that LENSR is able to learn high quality embeddings that are predictive of formula satisfiability. As a real-world case-study, we applied LENSR to the challenging task of Visual Relation Prediction (VRP) where the goal is to predict relations between objects in images. Our empirical analysis demonstrates that LENSR significantly outperforms baseline models. Furthermore, we observe that LENSR with d-DNNF achieves a significant performance improvement over LENSR with CNF embedding. We propose the notion of *embeddable-demanding* to capture the observed behavior of a plausible relationship between tractability of representation language and the ease of learning vector representations.

To summarize, this paper contributes a framework for utilizing logical formulae in NNs. Different from prior work, LENSR is able to utilize d-DNNF structure to learn semantically-constrained embeddings. To the best of our knowledge, this is also the first work to apply GCN-based embeddings for logical formulae, and experiments show the approach to be effective on both synthetic and real-world datasets. Practically, the model is straight-forward to implement and use. We have made our source code available online at `https://github.com/ZiweiXU/LENSR`. Finally, our evaluations suggest a connection between the tractability of a normal form and its amenability to embedding; exploring this relationship may reveal deep connections between knowledge compilation [14] and vector representation learning.

## 2   Background and Related Work

**Logical Formulae, CNF and d-DNNF**    Logical statements provide a flexible declarative language for expressing structured knowledge. In this work, we focus on *propositional logic*, where a *proposition* p is a statement which is either True or False. A *formula* F is a compound of propositions connected by logical connectives, e.g. $\neg, \wedge, \vee, \Rightarrow$. An *assignment* $\tau$ is a function which maps propositions to True or False. An assignment that makes a formula F True is said to *satisfy* F, denoted $\tau \models$ F.

A formula that is a conjunction of clauses (a disjunction of literals) is in Conjunctive Normal Form (CNF). Let $X$ be the set of propositional variables. A sentence in Negation Normal Form (NNF) is defined as a rooted directed acyclic graph (DAG) where each leaf node is labeled with True, False, $x$, or $\neg x, x \in X$; and each internal node is labeled with $\wedge$ or $\vee$ and can have arbitrarily many children. Deterministic Decomposable Negation Normal Form (d-DNNF) [13, 14] further imposes that the representation is: (i) **Deterministic:** An NNF is deterministic if the operands of $\vee$ in all well-formed boolean formula in NNF are mutually inconsistent; (ii) **Decomposable:** An NNF is

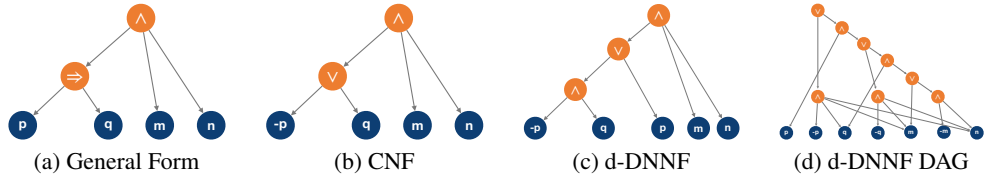

Figure 2: (a)–(c) Logic graphs examples of the formula $(p \implies q) \wedge m \wedge n$ in (a) General form, (b) CNF, (c) d-DNNF. This formula could encode a rule for "person wearing glasses" where p denotes wear(person,glasses), q denotes in(glasses,person), m denotes exist(person) and n denotes exist(glasses). (d) An example DAG showing a more complex d-DNNF logic rule.

decomposable if the operands of $\wedge$ in all well-formed boolean formula in the NNF are expressed on a mutually disjoint set of variables. In contrast to CNF and more general forms, d-DNNF has many desirable *tractability* properties (e.g., polytime satisfiability and polytime model counting). These tractability properties make d-DNNF particularly appealing for complex AI applications [16].

Although building d-DNNFs is a difficult problem in general, practical compilation can often be performed in reasonable time. We use c2d [17], which can compile relatively large d-DNNFs; in our experiments, it took less than 2 seconds to compile a d-DNNF from a CNF with 1000 clauses and 1000 propositions on a standard workstation. Our GCN can also embed other logic forms expressible as graphs and thus, other logic forms (e.g., CNF) could be used when d-DNNF compilation is not possible or prohibitive

**Logic in Neural Networks**   Integrating learning and reasoning remains a key problem in AI and encompasses various methods, including logic circuits [18], Logic Tensor Networks [19, 20], and knowledge distillation [5]. Our primary goal in this work is to incorporate symbolic domain knowledge into connectionist architectures. Recent work can be categorized into two general approaches.

The first approach augments the training objective with an additional logic loss as a means of applying soft-constraints [7, 21, 22, 23]. For example, the semantic loss used in [7] quantifies the probability of generating a satisfying assignment by randomly sampling from a predictive distribution. The second approach is via embeddings, i.e., learning vector based representations of symbolic knowledge that can be naturally handled by neural networks. For example, the ConvNet Encoder [8] embeds formulae (sequences of symbols) using a stack of one-dimensional convolutions. TreeRNN [9] and TreeLSTM encoders [12, 24, 11] recursively encode formulae using recurrent neural networks.

This work adopts the second embedding-based approach and adapts the Graph Convolutional Network (GCN) [15] towards embedding logical formulae expressed in d-DNNF. The prior work discussed above have focused largely on CNF (and more general forms), and have neglected d-DNNF despite its appealing properties. Unlike the ConvNet and TreeRNN/LSTM, our GCN is able to utilize semantic information inherent in the d-DNNF structure, while remaining invariant to proposition relabeling.

## 3   Logic Embedding Network with Semantic Regularization

In this section, we detail our approach, from logic graph creation to model training and eventual use on a target task. As a guide, Fig. 1 shows an overview of our model. LENSR specializes a GCN for d-DNNF formulae. A logical formula (and corresponding truth assignments) can be represented as a directed or undirected graph $\mathcal{G} = (\mathcal{V}, \mathcal{E})$ with $N$ nodes, $v_i \in \mathcal{V}$, and edges $(v_i, v_j) \in \mathcal{E}$. Individual nodes are either propositions (leaf nodes) or logical operators ($\wedge, \vee, \Rightarrow$), where subjects and objects are connected to their respective operators. In addition to the above nodes, we augment the graph with a global node, which is linked to all other nodes in the graph.

As a specific example (see Fig. 2), consider an image which contains a person and a pair of glasses. We wish to determine the relation between them, e.g., whether the person is wearing the glasses. We could use spatial logic to reason about this question; if the person is wearing the glasses, the image of the glasses should be "inside" the image of the person. Expressing this notion as a logical rule, we have: (wear(person,glasses) $\implies$ in(glasses, person)) $\wedge$ exist(person) $\wedge$ exist(glasses). Although the example rule above results in a tree structure, d-DNNF formulae are DAGs in general.

## 3.1 Logic Graph Embedder with Heterogeneous Nodes and Semantic Regularization

We embed logic graphs using a multi-layer Graph Convolutional Network [15], which is a first-order approximation of localized spectral filters on graphs [25, 26]. The layer-wise propagation rule is,

$$Z^{(l+1)} = \sigma \left( \tilde{D}^{-\frac{1}{2}} \tilde{A} \tilde{D}^{-\frac{1}{2}} Z^{(l)} W^{(l)} \right) \tag{1}$$

where $Z^{(l)}$ are the learnt latent node embeddings at $l^{th}$ (note that $Z^{(0)} = X$), $\tilde{A} = A + I_N$ is the adjacency matrix of the undirected graph $\mathcal{G}$ with added self-connections via the identity matrix $I_N$. $\tilde{D}$ is a diagonal degree matrix with $\tilde{D}_{ii} = \sum_j \tilde{A}_{ij}$. The layer-specific trainable weight matrices are $W^{(l)}$, and $\sigma(\cdot)$ denotes the activation function. To better capture the semantics associated with the logic graphs, we propose two modifications to the standard graph embedder: heterogenous node embeddings and semantic regularization.

**Heterogeneous Node Embedder.** In the default GCN embedder, all nodes share the same set of embedding parameters. However, different *types* of nodes have different semantics, e.g., compare an $\Rightarrow$ node v.s. a proposition node. Thus, learning may be improved by using distinct information propagation parameters for each node type. Here, we propose to use type-dependent logical gate weights and attributes, i.e., a different $W^{(l)}$ for each of the five node types (leaf, global, $\wedge$, $\vee$, $\Rightarrow$).

**Semantic Regularization.** d-DNNF logic graphs possess certain structural/semantic constraints, and we propose to incorporate these constraints into the embedding structure. More precisely, we regularize the children embeddings of $\wedge$ gates to be orthogonal. This intuitively corresponds to the constraint that the children do not share variables (i.e., $\wedge$ is decomposable). Likewise, we propose to constrain the $\vee$ gate children embeddings to sum up to a unit vector, which corresponds to the constraint that one and only one child of $\vee$ gate is true (i.e., $\vee$ is deterministic). The resultant semantic regularizer loss is:

$$\ell_r(\mathsf{F}) = \sum_{v_i \in \mathcal{N}_O} \left\| \sum_{\substack{\text{element-wise} \\ v_j \in \mathcal{C}_i}} q(v_j) - \mathbf{1} \right\|_2^2 + \sum_{v_k \in \mathcal{N}_A} \| V_k^T V_k - \text{diag}(V_k^T V_k) \|_2^2, \tag{2}$$

where $q$ is our logic embedder, $\mathcal{N}_O$ is the set of $\vee$ nodes, $\mathcal{N}_A$ is the set of $\wedge$ nodes, $\mathcal{C}_*$ is the set of child nodes of $v_*$, $V_k = [q(v_1), q(v_2), ..., q(v_l)]$ where $v_l \in \mathcal{C}_k$.

## 3.2 Embedder Training with a Triplet Loss

As previously mentioned, LENSR minimizes distances between the embeddings of formulae and satisfying assignments in a shared latent embedding space. To achieve this, we use a *triplet loss* that encourages formulae embeddings to be close to satisfying assignments, and far from unsatisfying assignments.

Formally, let $q(\cdot)$ be the embedding produced by the modified GCN embedder. Denote $q(\mathsf{F})$ as the embedding of d-DNNF logic graph for a given formula, and $q(\tau_\mathsf{T})$ and $q(\tau_\mathsf{F})$ as the assignment embeddings for a satisfying and unsatisfying assignment, respectively. For assignments, the logical graph structures are simple and shallow; assignments are a conjunction of propositions $\mathsf{p} \wedge \mathsf{q} \wedge \dots \mathsf{z}$ and thus, the pre-augmented graph is a tree with one $\wedge$ gate. Our triplet loss is a hinge loss:

$$\ell_t(\mathsf{F}, \tau_\mathsf{T}, \tau_\mathsf{F}) = \max\{d(q(\mathsf{F}), q(\tau_\mathsf{F})) - d(q(\mathsf{F}), q(\tau_\mathsf{T})) + m, 0\}, \tag{3}$$

where $d(x, y)$ is the squared Euclidean distance between vector $x$ and vector $y$, $m$ is the margin. We make use of SAT solver, `python-sat` [27], to obtain the satisfying and unsatisfying assignments. Training the embedder entails optimizing a combined loss:

$$L_{\text{emb}} = \sum_{\mathsf{F}} \sum_{\tau_\mathsf{T}, \tau_\mathsf{F}} \ell_t(\mathsf{F}, \tau_\mathsf{T}, \tau_\mathsf{F}) + \lambda_r \ell_r(\mathsf{F}), \tag{4}$$

where $\ell_t$ is the triplet loss above, $\ell_r$ is the semantic regularization term for d-DNNF formulae, and $\lambda_r$ is a hyperparameter that controls the strength of the regularization. The summation is over formulas and associated pairs of satisfying and unsatisfying assignments in our dataset. In practice, pairs of assignments are randomly sampled for each formula during training.

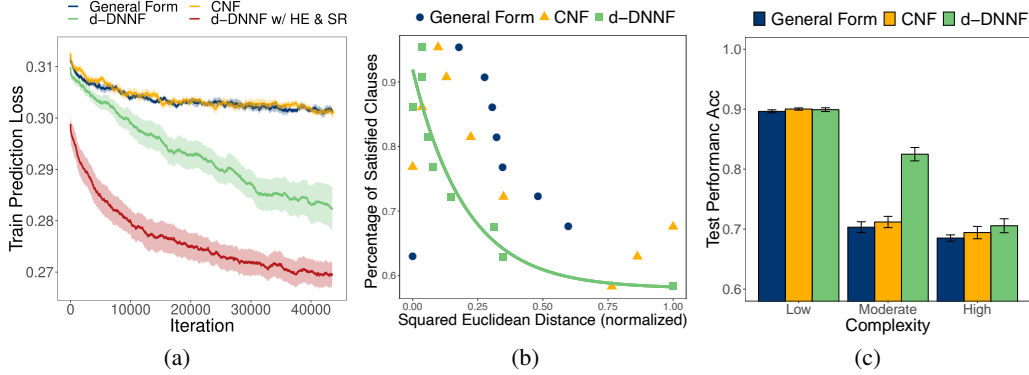

Figure 3: (a) Prediction loss (on the training set) as training progressed (line shown is the average over 10 runs with shaded region representing the standard error); (b) Formulae satisfiability v.s. distance in the embedding space, showing that LENSR learnt a good representation by projecting d-DNNF logic graphs; (c) Test accuracies indicate that the learned d-DNNF embeddings outperform the general form and CNF embeddings, and are more robust to increasing formula complexity.

### 3.3 Target Task Training with a Logic Loss

Finally, we train the target model $h$ by augmenting the per-datum loss with a logic loss $\ell_{\text{logic}}$ :

$$\ell = \ell_{\text{c}} + \lambda \ell_{\text{logic}}, \tag{5}$$

where $\ell_{\text{logic}} = \|q(\mathsf{F_x}) - q(h(x))\|_2^2$ is the embedding distance between the formula related to the input $x$ and the predictive distribution $h(x)$, $\ell_c$ is the task-specific loss (e.g., cross-entropy for classification), and $\lambda$ is a trade-off factor. Note that the distribution $h(x)$ may be any relevant predictive distribution produced by the network, including intermediate layers. As such, intermediate outputs can be regularized with prior knowledge for later downstream processing. To obtain the embedding of $h(x)$ as $q(h(x))$, we first compute an embedding for each predicted relationship $\mathsf{p}_i$ by taking an average of the relationship embeddings weighted by their predicted probabilities. Then, we construct a simple logic graph $G = \bigwedge_i \mathsf{p}_i$, which is embedded using q.

## 4 Empirical Results: Synthetic Dataset

In this section, we focus on validating that d-DNNF formulae embeddings are more informative relative to embeddings of general form and CNF formulae. Specifically, we conduct tests using a entailment checking problem; given the embedding of a formula f and the embedding of an assignment $\tau$, predict whether $\tau$ satisfies f.

**Experiment Setup and Datasets.** We trained 7 different models using general, CNF, and d-DNNF formulae (with and without heterogenous node embedding and semantic regularization). For this test, each LENSR model comprised 3 layers, with 50 hidden units per layer. LENSR produces 100-dimension embedding for each input formula/assignment. The neural network used for classification is a 2-layer perceptron with 150 hidden units. We set $m = 1.0$ in Eqn. 3 and $\lambda_r = 0.1$ in Eqn. 4. We used grid search to find reasonable parameters.

To explicitly control the complexity of formulae, we synthesized our own dataset. The complexity of a formula is (coarsely) reflected by its number of variables $n_v$ and the maximum formula depth $d_m$. We prepared three datasets with $(n_v, d_m) = (3, 3), (3, 6), (6, 6)$ and label their complexity as "low", "moderate", and "high". We attempted to provide a good coverage of potential problem difficulty: the "low" case represents easy problems that all the compared methods were expected to do well on, and the "high" case represents very challenging problems. For each formula, we use the python-sat package [27] to find its satisfying and unsatisfying assignments. There are 1000 formulae in each difficulty level. We take at most 5 satisfying assignments and 5 unsatisfying assignments for each formula in our dataset. We converted all formulae and assignments to CNF and d-DNNF.

Table 1: Prediction accuracy and standard error over 10 independent runs with model using different forms of formulae and regularization. Standard error shown in brackets. "HE" means the model is a heterogeneous embedder, "SR" means the model is trained with semantic regularization. "✓" denotes "with the respective property" and "-" denotes "Not Applicable". The best scores are in bold.

| Formula Form | HE | SR | Acc.(%) | | |
|---|---|---|---|---|---|
| | | | Low | Moderate | High |
| General | - | - | 89.63 (0.25) | 70.32 (0.89) | 68.51 (0.53) |
| CNF | | - | 90.02 (0.18) | 71.19 (0.93) | 69.42 (1.03) |
| | ✓ | - | 90.25 (0.15) | 73.92 (1.01) | 68.79 (0.69) |
| d-DNNF | | ✓ | 89.91 (0.31) | 82.49 (1.11) | 70.56 (1.16) |
| | | ✓ | 90.22 (0.23) | 82.28 (1.40) | 71.46 (1.17) |
| | ✓ | | 90.27 (0.55) | 81.30 (1.29) | 70.54 (0.62) |
| | ✓ | ✓ | **90.35** (0.32) | **83.04** (1.58) | **71.52** (0.54) |

**Results and Discussion.** Table 1 summarizes the classification accuracies across the models and datasets over 10 independent runs. In brief, the heterogeneous embedder with semantic regularization trained on d-DNNF formulae outperforms the alternatives. We see that semantic regularization works best when paired with heterogeneous node embedding; this is relatively unsurprising since the AND and OR operators are regularized differently and distinct sets of parameters are required to propagate relevant information.

In our experiments, we found the d-DNNF model to converge faster than the CNF and general form (Fig. 3a). Utilizing both semantic regularization and heterogeneous node embedding further improves the convergence rate. The resultant embedding spaces are also more informative of satisfiability; Fig. 3b shows that the distances between the formulae and associated assignments better reflect satisfiability for d-DNNF. This results in higher accuracies (Fig. 3c), particularly on the moderate complexity dataset. We posit that the differences on the low and high regimes were smaller because (i) in the low case, all the methods performed reasonably well, and (ii) on the high regime, embedding the constraints helps to a limited extent and points to avenues for future work.

Overall, these results provide empirical evidence for our conjecture that d-DNNF are more amenable to embedding, compared to CNF and general form formulae.

## 5  Visual Relation Prediction

In this section, we show how our framework can be applied to a real-world task — Visual Relation Prediction (VRP) — to train improved models that are consistent with both training data and prior knowledge. The goal of VRP is to predict the correct relation between two objects given visual information in an input image. We evaluate our method on VRD [28]. The VRD dataset contains 5,000 images with 100 object categories and 70 annotated predicates (relations). For each image, we sample pairs of objects and induce their spatial relations. If there is no annotation for a pair of object in the dataset, we label it as having "no-relation".

**Propositions and Constraints** The logical rules for the VRP task consist of logical formulae specifying constraints. In particular, there are three types of propositions in our model:

- **Existence Propositions** The existence of each object forms a proposition which is True if it exists in the image and False otherwise. For example, proposition p=exist(person) is True if a person is in the input image and False otherwise.

- **Visual Relation Propositions** Each of the candidate visual relation together with its subject and object forms a proposition. For example, wear(person, glasses) is a proposition and has value True if there is a person wearing glasses in the image and False otherwise.

- **Spatial Relation Propositions** In order to add spatial constraints, e.g. a person cannot wear the glasses if their bounding boxes do not overlap, we define 10 types of spatial relationships (illustrated in Fig. 4a). We assign a proposition for each spatial relation such that the proposition evaluation is True if the relation holds and False otherwise, e.g. in(glasses, person). Furthermore, exactly one spatial relation proposition for a fixed subject

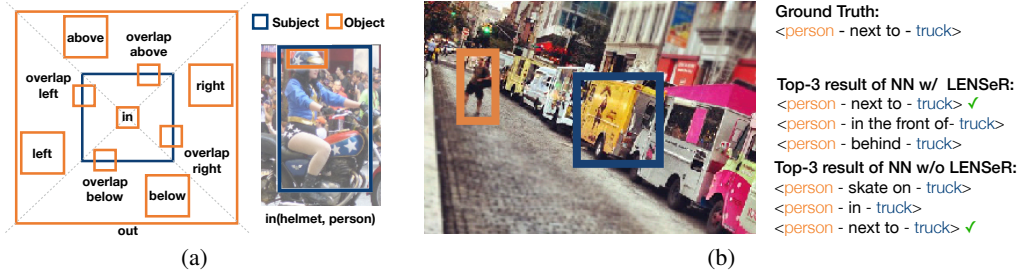

(a)                                                    (b)

Figure 4: (a) The 10 spatial relations used in the Visual Relation Prediction task, and an example image illustrating the relation in(helmet, person). (b) A prediction comparison between neural networks trained w/ and w/o LENSR. A tick indicates a correct prediction. In this example, the misleading effects of the street are corrected by spatial constraints on "skate on".

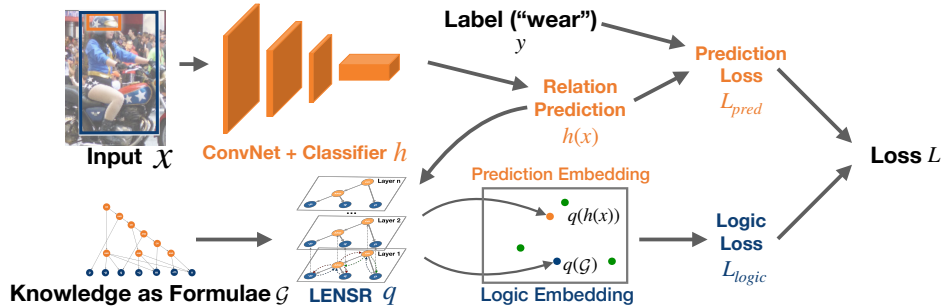

Figure 5: The framework we use to train VRP models with LENSR.

object pair is True, i.e. spatial relation propositions for a fixed subject object pair are mutually exclusive.

The above propositions are used to form two types of logical constraints:

- **Existence Constraints.** The prerequisite of any relation is that relevant objects exist in the image. Therefore p(sub,obj) $\implies$ (exist(sub) $\wedge$ exist(obj)), where p is any of the visual or spatial relations introduced above.

- **Spatial Constraints.** Many visual relations hold only if a given subject and object follow a spatial constraint. For example, a person cannot be wearing glasses if the bounding boxes for the person and the glasses do not overlap. This observation gives us rules such as wear(person, glasses) $\implies$ in(glasses, person).

For each image $i$ in the training set we can generate a set of clauses $F_i = \{c_{ij}\}$ where $c_{ij} = \bigvee_k \mathsf{p}_{jk}$. Each clause $c_{ij}$ represents a constraint in image $i$ where $j$ is the constraint index, and each proposition $\mathsf{p}_{jk}$ represents a relation in image $i$ where $k$ is the proposition index in the constraint $c_{ij}$. We obtain the relations directly from the annotations for the image and calculate the constraints based on the definitions above. Depending on the number of objects in image $i$, $F_i$ can contain 50 to 1000 clauses and variables. All these clauses are combined together to form a formula $\mathsf{F} = \bigwedge_i F_i$.

## 5.1 VRP Model Training

Using the above formulae, we train our VRP model in a two-step manner; first, we train the embedder $q$ using only f. The embedder is a GCN with the same structure as described in Sec. 4. Then, the embedder is fixed and the target neural network $h$ is trained to predict relation together with the logic loss (described in Sec. 3.3). The training framework is illustrated in Fig. 5. In our experiment, $h$ is a MLP with 2 layers and 512 hidden units. To elaborate:

**Embedder Training.** For each training image, we generate an intermediate formula $\mathsf{f}_i$ that only contains propositions related to the current image $I$. To do so, we iterate over all clauses in f and add

a clause $c_i$ into the intermediate formula if all subjects and objects of all literals in $c_i$ is in the image. The formula $\mathsf{f}_i$ is then appended with existence and spatial constraints defined in Sec. 5.

To obtain the vector representation of a proposition, we first convert its corresponding relation into a phrase (e.g. $\mathsf{p}$=wear(person, glasses) is converted to "person wear glasses"). Then, the GLoVe embeddings [29] for each word are summed to form the embedding for the entire phrase. The formula is then either kept as CNF or converted to d-DNNF [17] depending on the embedder. Similar to Sec. 4, the assignments of $\mathsf{f}_i$ are found and used to train the embedder using the triplet loss (Eqn. 3).

**Target Model Training.** After $q$ is trained, we fix its parameters and use it to train the relation prediction network $h$. In our relation prediction task, we assume the objects in the images are known; we are given the object labels and bounding boxes. Although this is a strong assumption, object detection is an upstream task that is handled by other methods and is not the focus of this work. Indeed, all compared approaches are provided with *exactly the same information*. The input to the model is the image, and the labels and bounding boxes of all detected objects, for example: $(I, [(\mathsf{table}, [23, 78, 45, 109]), (\mathsf{cup}, [10, 25, 22, 50])])$. The network predicts a relation based on the visual feature and the embedding of class labels:

$$\hat{y} = h([r_1, b_1, r_2, b_2, v]), \tag{6}$$

where $\hat{y}$ is the relation prediction, $r_i = \mathrm{GLoVe}(\mathrm{label}_i)$ is the GLoVe embedding for the class labels of subjects and objects, $b_i$ is the relative bounding box positions of subjects and objects, $v = \mathrm{ResNet}(I_{bbox_1 \cup bbox_2})$ is the visual feature extracted from the union bounding box of the objects, $[\cdot]$ indicates concatenation of vectors. We compute the logic loss term as

$$L_{\mathrm{logic}} = \left\| q(\mathsf{f}) - q\Big( \bigwedge_i \mathsf{p}_i \Big) \right\|_2^2, \tag{7}$$

where $\mathsf{p}_i$ is the predicate for $i^{\mathrm{th}}$ relation predicted to be hold in the image, and $\mathsf{f}$ is the formula generated from the input information. As previously stated, our final objective function is $L = L_{\mathrm{c}} + \lambda L_{\mathrm{logic}}$ where $L_c$ is the cross entropy loss, and $\lambda = 0.1$ is a trade-off factor. We optimized this objective using Adam [30] with learning rate $10^{-3}$.

Although our framework can be trained end-to-end, we trained the logic embedder and target network separately to (i) alleviate potential loss fluctuations in joint optimization, and (ii) enable the same logic embeddings to be used with different target networks (for different tasks). The networks could be further optimized jointly to fine-tune the embeddings for a specific task, but we did not perform fine-tuning for this experiment.

## 5.2 Empirical Results

Table 2 summarizes our results and shows the top-5 accuracy score of the compared methods[2]. We clearly see that our GCN approach (with heterogeneous node embedding and semantic regularization) performs far better than the baseline model without logic embeddings. Note also that direct application of d-DNNFs via the semantic loss [7] only resulted in marginal improvement over the baseline. A potential reason is that the constraints in VRP are more complicated than those explored in prior work: there are thousands of propositions and a straightforward use of d-DNNFs causes the semantic loss to rapidly approach $\infty$. Our embedding approach avoids this issue and thus, is able to better leverage the encoded prior knowledge. Our method also outperforms the state-of-the-art TreeLSTM embedder [12]; since RNN-based embedders are not invariant to variable-ordering, they may be less appropriate for symbolic expressions, especially propositional logic.

As a qualitative comparison, Fig. 4b shows an example where logic rules embedded by LENSR help the target task model. The top-3 predictions of neural network trained with LENSR are all reasonable answers for the input image. However, the top 3 relations predicted baseline model are unsatisfying and the model appears misled by the street between the subject and the object. LENSR leverages the logic rules that indicate that the "skate on" relation requires the subject to be "above" or "overlap above" the object, which corrects for the effect of the street.

Table 2: Performance of VRP under different configurations. "HE" indicates a heterogeneous node embedder, "SR" means the model uses semantic regularization. "✓" denotes "with the respective property" and "-" denotes "Not Applicable". The best scores are in bold.

| Model | Form | HE | SR | Top-5 Acc. (%) |
|---|---|---|---|---|
| without logic | - | - | - | 84.30 |
| with semantic loss [7] | - | - | - | 84.76 |
| with treeLSTM embedder [12] | CNF | - | - | 85.76 |
| | d-DNNF | - | - | 82.99 |
| LENSR | CNF | | - | 85.39 |
| | | ✓ | - | 85.70 |
| | d-DNNF | | ✓ | 85.37 |
| | | | ✓ | 88.01 |
| | | ✓ | | 90.13 |
| | | ✓ | ✓ | **92.77** |

## 6 Discussion

Our experimental results show an interesting phenomena—the usage of d-DNNF (when paired with semantic regularization) significantly improved performance compared to other forms. This raises a natural question of whether d-DNNF's embeddings are easier to learn. Establishing a complete formal connection between improved learning and compiled forms is beyond the scope of this work. However, we use the size of space of the formulas as a way to argue about ease of learning and formalize this through the concept of embeddable demanding.

**Definition 1 (Embeddable-Demanding)** *Let $L_1, L_2$ be two compilation languages. $L_1$ is at least as embeddable-demanding as $L_2$ iif there exists a polynomial $p$ such that for every sentence $\alpha \in L_2, \exists \beta \in L_1$ such that (i) $|\beta| \leq p(|\alpha|)$. Here $|\alpha|, |\beta|$ are the sizes of $\alpha, \beta$ respectively, and $\beta$ may include auxiliary variables. (ii) The transformation from $\alpha$ to $\beta$ is poly time. (iii) There exists a bijection between models of $\beta$ and models of $\alpha$.*

**Theorem 1** *CNF is at least as embeddable-demanding as d-DNNF, but if d-DNNF is at least as embeddable-demanding as CNF then $P = PP$.*

The proof and detailed theorem statement are provided in the Appendix. More broadly, Theorem 1 a first-step towards a more comprehensive theory of the embeddability for different logical forms. Future work in this area could potentially yield interesting insights and new ways of leveraging symbolic knowledge in deep neural networks.

## 7 Conclusion

To summarize, this paper proposed LENSR, a novel framework for leveraging prior symbolic knowledge. By embedding d-DNNF formulae using an augmented GCN, LENSR boosts the performance of deep NNs on model-checking and VRP tasks. The empirical results indicate that constraining embeddings to be semantically faithful, e.g., by allowing for node heterogeneity and through regularization, aids model training. Our work also suggests potential future benefits from a deeper examination of the relationship between tractability and embedding, and the extension of semantic-aware embedding to alternative graph structures. Future extensions of LENSR that embed other forms of prior symbolic knowledge could enhance deep learning where data is relatively scarce (e.g., real-world interactions with humans [31] or objects [32]). To encourage further development, we have made our source code available online at `https://github.com/ZiweiXU/LENSR`.

**Acknowledgments**

This work was supported in part by a MOE Tier 1 Grant to Harold Soh and by the National Research Foundation Singapore under its AI Singapore Programme [AISG-RP-2018-005]. It was also supported by the National Research Foundation, Prime Minister's Office, Singapore under its Strategic Capability Research Centres Funding Initiative.

## Footnotes

*Equal contribution and the rest of the authors are ordered alphabetically by last name.

[2]Top-5 accuracy was used as our performance metric because a given pair of objects may have multiple relationships, and reasonable relations may not have been annotated in the dataset.

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
