[Supplementary Material]

# 1   Supplementary Material

## 1.1   Embedder Training Algorithm

The algorithm for training the embedder is summarized in Algorithm 1.

---

**Algorithm 1 trainEmbedder**

---

**Input**: f: CNF formula; $I$: Training images; $m$: Margin
**Output**: $q$: Embedder

```
 1: q ← init()
 2: repeat
 3:    for all I_i ∈ I do
 4:       // Create intermediate formula
 5:       f_i ← { }
 6:       objs ← all_instances_in(I_i)
 7:       for all c_i ∈ f do
 8:          if all_instances_in(c_i) ∈ objs then
 9:             f_i ← f_i ∪ c_i
10:          end if
11:       end for
12:       f_i ← append_constraints(f_i)
13:       τ_T ← sat_assig_of(f_i)
14:       τ_F ← unsat_assig_of(f_i)
15:       Θ_q ← argmin L_emb
                   Θ_q
16:       q ← update_with(q,Θ_q)
17:    end for
18: until Convergence
19: return  q
```

---

## 1.2   Embeddable-Demanding

The significant performance improved due to usage of d-DNNF raises the question whether the language represented by d-DNNF has a smaller search space and therefore, potentially easier learning method. To this end, we introduce the concept of embeddable-demanding below

The following theorems uses the standard complexity theoretic terms and we refer to the reader to the standard text [1] for detailed treatment of these concepts.

**Definition 1 (Embeddable-Demanding)** *Let $L_1, L_2$ be two compilation languages. $L_1$ is at least as embeddable-demanding as $L_2$ iff there exists a polynomial $p$ such that for every sentence $\alpha \in L_2, \exists \beta \in L_1$ such that (i) $|\beta| \leq p(|\alpha|)$. Here $|\alpha|, |\beta|$ are the sizes of $\alpha, \beta$ respectively, and $\beta$ may include auxiliary variables. (ii) The transformation from $\alpha$ to $\beta$ is poly time. (iii) There exists a bijection between models of $\beta$ and models of $\alpha$.*

**Theorem 1.1** *CNF is at least as embeddable-demanding as d-DNNF but if d-DNNF is at least as embeddable-demanding as CNF then $P = PP$*

**Proof 1.1** *(1) Prove that CNF is at least as embeddable-demanding as d-DNNF, i.e. for every formula $\alpha$ in d-DNNF, there exists a polynomial size, and polynomial time computable CNF formula $\beta$ such that there is an one to one polynomial time computable mapping between models of $\beta$ to $\alpha$.*

*Observe that d-DNNF represents a circuit, which can be encoded into an equisatisfiable CNF formula of polynomial size due to NP-completeness of CNF. In particular, the usage of Tseytin encoding [2] ensures that the resulting CNF is of linear size. Furthermore, let d-DNNF $G$ be defined over the set of variables denoted by $X$, then Tseytin encoding introduces a set of auxiliary variables, say $Y$, for the resulting formula $F$ such that $G(X) = \exists Y F(X \cup Y)$. Therefore, the mapping from models of $G$ to $F$ is achieved just by projection of models of $G$ on $X$.*

*(2) Prove that if d-DNNF is at least as embeddable-demanding as CNF then $P = PP$. In other words, if for every formula $\beta$ in CNF, there exists a polynomial size, and polynomial time computable d-DNNF $\alpha$ such that there is bijection between models of $\alpha$ and models of $\beta$, then $P = PP$. $P = PP$ implies collapse of entire polynomial hierarchy, in particular $P = NP$.*

*Assume for every formula $\beta$ in CNF, there exists a polynomial size, and polynomial time computable d-DNNF $\alpha$ such that there is a bijection between models of $\alpha$ and models of $\beta$. Since d-DNNF allows counting in polynomial time and the existence of bijection implies that the number of models of $\alpha$ is equal to that of $\beta$, then we can compute the number of models of an arbitrary CNF formula in polynomial time; therefore $P = PP$. In this context, it is worth noting that the entire polynomial polynomial hierarchy is shown to contain $PP$, i.e., $PH \subseteq PP$ [3].*

## 1.3  Computing Infrastructure

We trained our models using `Pytorch` 0.4.1 on one NVIDIA GTX 1080 Ti 12GB GPU.

## 1.4  Hyper-parameters Selection

Our hyper-parameters includes: the margin in triplet loss of the embedder $m$, the semantic regularizer weight $\lambda_r$ and logic loss weight $\lambda$. The ranges considered are $[0.5, 5]$ for $m$; $[0.05, 0.2]$ for $\lambda_r$ and $[0.05, 0.2]$ for $\lambda$. We did grid search and set $m = 1.0, \lambda_r = 0.1, \lambda = 0.1$ across all experiments.