[Reviews · NeurIPS 2019]

Reviewer 1



This paper clearly build on a foundation of previous work; but it's approach appears original to me. I want to commend the authors for writing a very clear and easily readable paper. I also like that the discussion includes an attempt to provide intuition for why the CNF approach does not work as well as the d-DNNF. Nice. The quality of the paper is good. The authors do a thorough job explaining their approach and validating it with relevant data sets. One minor complaint is that although Fig 3c showed very good improvement for moderate complexity, the high complexity case is not that compelling. The authors shouldn't point the the high case as evidence. In fact, I'd like to see more discussion of why the low and high cases don't show as much improvement as the moderate case. That could lead to some deeper understanding of the method. This topic is very relevant and significant to many NeurIPS attendees, and will continue to be for the foreseeable future.

Reviewer 2



This paper introduces a method for incorporating prior knowledge encoded as logical rules to improve the performance of deep learning models. In particular, it takes logical rules which are in decomposable and deterministic negation normal form (d-DNNF), and proposes using an augmented graph convolution network to embed them into a vector space. This embedding is then regularised according to the logical constraints, allowing the addition of a "logic loss" term to train models obeying these logical rules. Incorporating (symbolic) background knowledge to improve performance of deep learning methods is an interesting and valuable direction, and from the experiments using a d-DNNF rather than a CNF appears to be beneficial. However, for me the notion of using a d-DNNF as the source of background knowledge raises a few issues which I feel are not addressed in the paper. In general, building d-DNNFs is a difficult problem - taking an arbitrary logical formula and compiling a d-DNNF from it is implicitly solving the #P-hard problem of doing model counting on the formula (because d-DNNFs are a "tractable form" and can do many difficult computations easily). This problem has been studied extensively ("A Compiler for Deterministic, Decomposable Negation Normal Form, Darwiche 2002", "New Advances in Compiling CNF into Decomposable Negation Normal Form, Darwiche 2004", "A Top-Down Compiler for Sentential Decision Diagrams. Oztok and Darwiche 2015" for a particular class of d-DNNFs), but this difficulty is completely hidden in the paper, and rather taken as a given that we can easily construct a d-DNNF. In addition to being difficult to compile, having a built d-DNNF allows you to solve difficult problems easily, which brings me to my second concern. The experiments compare the results of using d-DNNFs vs. CNFs as the underlying embedding structure, but do not compare a more direct approach. That is, you could take something like semantic loss ("A Semantic Loss Function for Deep Learning with Symbolic Knowledge, Xu et al 2018"), and use the d-DNNF directly in the loss to enforce constraints without embedding the formula and having the inaccuracies that arise (contrary to the description in the paper, semantic loss requires solving exactly the same problem as this paper - building a d-DNNF, and does not require any sampling or approximation). To me this is the more meaningful comparison, since if the d-DNNF is already compiled there must be a justification for why the logical embedding is needed at all. The paper is mostly quite clear: the technical content was understandable with the exception that I found it a bit difficult to understand exactly which part of the network these constraints addressed. This was mentioned later, but more explicitly clarifying it early on would be helpful. The setup and result of the experiments were easy to analyze, and I like that the authors used a real task and data set to validate their approach. I found the discussion to be a little bit confusing - it seems like the claim is that embedding CNFs is harder because any d-DNNF can be written as a similar sized CNF (via tseytin encoding), but not every d-DNNF can be written as a similar sized CNF (due to weighted model counting being PP-hard), and therefore d-DNNFs may be easier to learn? While correct, this formalization does not make it any easier to understand, and more intuition is definitely required. The idea of why embedding d-DNNFs is more effective than CNFs is an interesting point to discuss (even if the discussion is inconclusive), but it could be made more intuitive, and also perhaps mention the inherent structure that is already in d-DNNFs, rather than just the sizes of the search spaces. Post-rebuttal: The paper as is still has some motivational issues for me regarding the use of d-DNNFs, but I feel confident the authors can produce a good camera ready, which I would expect to include further discussion of the use of other embeddings and the effects on the technique (with some experimental ablations regarding the effect glove has), as well as a clearer motivation throughout the paper of why having a technique like this is important as compared to using d-DNNFs as is.

Reviewer 3



This paper presents a novel idea to integrate logical knowledge with deep neural networks via logic graph embeddings. The proposed model aims at converting logic formulas to d-DNNF forms such that GCN can be utilized to learn a single embedding for each formula that could be trained through back-propagation. The logic embedder is then further used to improve the main task by incorporating a logic loss. Overall, this paper is well-written. The proposed methodology is novel and interesting. The authors conduct extensive experiments on both synthetic and real-world datasets to demonstrate the advantage of the proposed model. It is clear that d-DNNF is preferred compared to other logical forms (e.g., general and CNF) from the experimental results. However, it is not too clear to me how exactly the model can be implemented from the descriptions. The detailed comments are the following: 1. Section 3.1 states that there are 5 types of nodes, but "IMPLY" is replaced with disjunctions in both CNF and d-DNNF forms. Should there be only 4 node types for these situations? 2.In section 3.2, how do you obtain the assignment embedding, e.g., p^q^...z? Do you initialize the embeddings for p,q,...z and then train them jointly? Do you treat the embedding of the topmost node as the formula embedding? Then what's the use of the additional global node mentioned in line 93? More details should be given in section 3.2 to illustrate how to compute q(f) and q(F/T). 3. In section 3.3, it is also unclear of how to compute q(h). Given a neural network, how does it contribute to the logic embeddings, i.e., what does those p_i correspond in this mapping? 4. The model separates the training of a logic embedder and the final target task. I'm wondering why not combine them jointly so that training loss can be propagated and the framework becomes end-to-end? 5. For experiments, what's the statistics of the synthetic dataset, besides the 3 different levels. (What's the number of training instances for each dataset?) For Figure 3(c), the performance pattern seems not normal. The proposed model has large gains for moderate dataset but less obvious in high dataset. Can you explain the reason? Moreover, the results for both synthetic dataset and VRP dataset are not too convincing when the authors only compare with different logic forms and treeLSTM. There are many neuro-symbolic methods, e.g., Logic Tensor Networks. It could be better to compare with those architectures.

[Author Response · NeurIPS 2019]

Thank you to the reviewers for their detailed comments. We commit to addressing the minor and typographical errors. In the following, we discuss the main issues raised:

**1. (R2) "if the d-DNNF is already compiled there must be a justification for why the logical embedding is needed"** and **"compare a more direct approach... like semantic loss (Xu et al 2018), and use the d-DNNF directly in the loss to enforce constraints without embedding the formula."**

**Re:** As suggested, we compared our method against using d-DNNFs directly via the semantic loss proposed in (Xu et al. 2018)[1] on the visual relation prediction (VRP) task. Our method outperforms the semantic loss (Table 1).

Table 1: Comparison with semantic loss on visual relation prediction (VRP) task.

| Model | Top-5 Acc.(%) |
|---|---|
| w/o semantic loss and w/o our embedder | 84.30% |
| with semantic loss | 84.76% |
| with GCN embeddings (ours) | **92.77%** |

A potential reason for the performance difference is that the constraints in VRP are more complicated than those explored in (Xu et al., 2018): there are thousands of propositions and a direct use of d-DNNFs causes the semantic loss to rapidly approach $\infty$. In contrast, our embedding approach avoids this issue. Moreover, our approach enables generalization to constraints that involve previously unseen propositions; we can leverage representations such as GLoVE word embeddings, which is not possible in the semantic loss. We will add these results and discussion to the paper.

**2. (R2) "Building d-DNNFs is a difficult problem ... this difficulty is completely hidden in the paper"**
**Re:** Indeed, building d-DNNFs is a difficult problem in general and we will make this fact clearer in the paper. Practical compilation has progressed significantly thanks to research in the area (e.g., the prior work pointed out by R2). We use `c2d` (Darwiche, 2004) that can compile relatively large d-DNNFs in reasonable time; in our experiments, it took less than 2 seconds to compile a d-DNNF from a CNF with 1000 clauses and 1000 propositions on a standard workstation.

We would like to emphasize that our GCN can embed other logic forms expressible as graphs. For cases where d-DNNF compilation is not possible or prohibitive, other logic forms (e.g., CNF) could be used. The key contribution of our approach is improved accuracy by embedding logical constraints. As suggested by the reviewer, we will highlight the trade-off between accuracy and cost (to obtain compiled forms) in the final version.

**3. (R3) "Why not combine them jointly so that ... the framework becomes end-to-end?"**
Our framework can be trained end-to-end. However, we train the networks separately to (i) alleviate potential loss fluctuations in joint optimization which makes training easier, and (ii) enable the same logical embeddings to be used with different target networks (for different tasks). As suggested by the reviewer, the networks could be trained end-to-end to fine-tune the embeddings further for a specific task.

**4. (R1&R3) "... more discussion of why the low and high cases don't show as much improvement as the moderate case"**
**Re:** We will provide additional discussion in the revision. In brief, we attempted to provide a good coverage of potential problem difficulty. The "low" case represents easy problems that all the compared methods were expected to do well on, and the "high" case represents very challenging problems that all methods were expected to struggle on. We posit that the difference on the low and high regimes were smaller because (i) in the low case, all the methods performed reasonably well, and (ii) on the high regime, embedding the constraints helps to a limited extent and points to avenues for future work.

**5. (R3) "Comparison with some state-of-the-art neuro-symbolic methods, e.g. Logic Tensor Networks, could be a plus"**
**Re:** Neuro-symbolic methods, such as Logic Tensor Network (LTN) and logic circuits, are not directly applicable in our setting since the constraints for each input may differ. For example, in the visual relation task, the constraints for each image was different. One possible approach would be to conjunct all the constraints, but there would be $7 \times 10^5$ propositions and $\approx 10^5$ clauses, which results in a prohibitively large logic network.

**6. (R3) "Provide more details on the methodology such that it could be better understood and enhance its reproducibility."** and **"explain how they chose some of their parameters."**
**Re:** We used grid search to find reasonable parameters; we will state this and provide additional methodological details. We have also submitted our implementation code which can be used to reproduce all the results in the paper.

## Footnotes

[1] We used publicly available code provided by the authors.


[Meta-Review · NeurIPS 2019]

The paper describes a novel way of regularizing a deep neural network to be semantically similar to a logical formula. The main contribution is the use of d-DDNF, a particular format for formula which is well-suited to embedding in a graph CNN. This format is used in certain reasoning tasks, but is not widely known in the NeurIPS community, and (to this metareviewer at least) seeing it in this context was surprising and insightful. It also seems to be a trick that could be useful across a range of tasks. It is shown that the model improves performance on synthetic data and a non-trivial realistic task, visual relation prediction. The paper is well-written, and judged to contain results of some significance by all reviewers, and one of the reviewers also rated the paper as a strong accept.